# Soil Bacterial Community Characteristics and Functional Analysis of Estuarine Wetlands and Nearshore Estuarine Wetlands in Qinghai Lake

**DOI:** 10.3390/microorganisms13040759

**Published:** 2025-03-27

**Authors:** Wei Ji, Zhiyun Zhou, Jianpeng Yang, Ni Zhang, Ziwei Yang, Kelong Chen, Yangong Du

**Affiliations:** 1Qinghai Province Key Laboratory of Physical Geography and Environmental Process, College of Geographical Science, Qinghai Normal University, Xining 810008, China; jiwei100500@163.com (W.J.); 13897423633@163.com (Z.Z.); 13079313886@163.com (J.Y.); zhangni0224@163.com (N.Z.); 15756789182@163.com (Z.Y.); 2Lianyungang Academy of Agricultural Sciences, Lianyungang 222006, China; 3Key Laboratory of Surface Processes and Ecological Conservation on the Tibetan Plateau, Qinghai Normal University, Xining 810008, China; 4National Positioning Observation and Research Station of Qinghai Lake Wetland Ecosystem in Qinghai, National Forestry and Grassland Administration, Haibei 812300, China; 5Northwest Institute of Plateau Biology, Chinese Academy of Sciences, Xining 810008, China

**Keywords:** Qinghai Lake basin, estuarine wetlands, nearshore estuarine wetlands, bacterial community, functional analysis

## Abstract

Qinghai Lake, the largest inland saline lake in China, plays a vital role in wetland carbon cycling. However, the structure and function of soil bacterial communities in its estuarine and nearshore estuarine wetlands remain unclear. This study examined the effects of wetland type and soil depth on bacterial diversity, community composition, and functional potential in the Shaliu, Heima, and Daotang River wetlands using high-throughput sequencing. The results showed that wetland type and soil depth significantly influenced bacterial communities. Nearshore wetlands exhibited lower bacterial diversity in the 0–10 cm layer, while deeper soils (10–20 cm) showed greater regional differentiation. Estuarine wetlands were enriched with Proteobacteria, Actinobacteriota, and Chloroflexi, whereas nearshore wetlands were dominated by Actinobacteriota and Cyanobacteria. Functionally, estuarine wetlands had higher sulfate reduction and anaerobic decomposition potential, with Desulfovibrio, Desulfobacter, and Desulfotomaculum regulating sulfur cycling and carbon decomposition. In contrast, nearshore wetlands showed greater nitrogen fixation and organic matter degradation, facilitated by Rhizobium, Azotobacter, Clostridium, and nitrogen-fixing Cyanobacteria (e.g., Anabaena, Nostoc). Microbial metabolic functions varied by depth: surface soils (0–10 cm) favored environmental adaptation and organic degradation, whereas deeper soils (10–20 cm) exhibited lipid metabolism and DNA repair strategies for low-oxygen adaptation. These findings highlight the spatial heterogeneity of bacterial communities and their role in biogeochemical cycles, providing insights into wetland carbon dynamics and informing conservation strategies.

## 1. Introduction

Microbial communities, formed by the interactions of various microbial populations within specific habitats, are fundamental structural units that play a crucial role in ecosystem functions and material cycling [1]. As the core components of soil ecosystems, soil microorganisms—including bacteria, fungi, archaea, and protozoa [2]—are not only highly diverse and abundant, but also play a pivotal role in biogeochemical cycles and energy flows within the biosphere, atmosphere, and pedosphere [3]. By participating in elemental cycling, organic matter decomposition, and degradation processes, these microorganisms maintain and enhance soil nutrients, fertility, and productivity. At the same time, they exhibit high sensitivity to changes in soil environments and anthropogenic disturbances [4,5]. Consequently, soil microbial indicators are widely used in soil quality assessments, offering a more comprehensive reflection of soil health compared to traditional physical and chemical indicators [6]. Studies have shown that soil microbial diversity is closely related to ecosystem functions and environmental adaptability. Its diversity indices hold significant potential for assessing ecosystem functionality [7,8,9].

Wetlands, with their high primary productivity and biodiversity, are vital components of the global carbon cycle [10,11,12]. Wetlands serve not only as significant sources of greenhouse gases, but also as remarkable carbon sinks through carbon fixation and storage, playing an irreplaceable role in addressing global climate change [13,14,15]. Among various wetland types, estuarine wetlands account for only 3.4% of the global wetland area but exhibit an impressive annual carbon accumulation rate of 250–500 g/m^2^, highlighting their substantial carbon sink potential [16,17]. In wetland soils, carbon primarily exists in the form of undecomposed plant litter, significantly influenced by soil microbial community activity [18,19]. Studies have demonstrated that soil organic matter (SOM) contains a substantial amount of microbially derived carbon, whose stability depends on microbial activity and its interaction with minerals [20,21]. Moreover, the “microbial carbon pump” (MCP) theory provides additional insight, revealing that microbial cell components and degradation products generated during rapid microbial turnover can be preserved over the long term, offering a theoretical basis for wetland carbon storage [22]. Microorganisms play multifaceted roles in the wetland carbon cycle. On the one hand, they reduce carbon storage by releasing CO₂ and CH₄ through decomposition metabolism [23,24,25]. On the other hand, they enhance carbon storage by transforming labile carbon into more stable organic matter through biosynthetic processes [26,27].

Qinghai Lake, the largest inland saline lake in China, is surrounded by estuarine and nearshore wetlands that are critical regions for studying wetland ecosystem functions and carbon cycling. However, research on soil microbial communities in these wetlands remains relatively limited. This study aims to analyze soil bacterial diversity and community structure in the estuarine and nearshore wetlands of Qinghai Lake, exploring the functional characteristics of the lake’s wetland ecosystems. The findings are intended to provide a scientific basis for assessing and managing wetland ecosystem functions.

## 2. Materials and Methods

### 2.1. Study Area Overview

This study focuses on the estuarine wetlands and nearshore estuarine wetlands (approximately 200 m from the estuarine wetlands) of Qinghai Lake. Six experimental sites were established: the nearshore estuarine wetland (SC1) and estuarine wetland (SC2) in the Shaliu River basin, the nearshore estuarine wetland (HC1) and estuarine wetland (HC2) in the Heima River basin, and the nearshore estuarine wetland (DC1) and estuarine wetland (DC2) in the Daotang River basin. Each site was set up with three replicates, with detailed locations shown in Figure 1.

The study area covers the estuarine wetlands of the Shaliu River (37°12′ N, 100°11′ E; elevation of 3197–3199 m), Heima River (36°43′ N, 99°47′ E; elevation of 3197–3201 m), and Daotang River (36°34′ N, 100°44′ E; elevation of 3201–3203 m). All these regions are located in the plateau continental climate zone, with soils primarily consisting of marsh soils and meadow soils. The dominant vegetation includes *Eleusine indica*, *Cyperus rotundus*, *Argentina anserina*, *Sibbaldianthe bifurca*, and *Cymbopogon citratus*, which collectively represent the typical ecological characteristics of the three river basin wetlands.

### 2.2. Sample Collection

Soil samples were collected and processed in early August 2023. Using a soil auger with a diameter of 4.5 cm, samples were taken at each plot following the five-point sampling method at two depths: 0–10 cm (a) and 10–20 cm (b). After collection, stones and plant debris were manually removed, and the soil was homogenized and passed through a 2 mm sieve. The sieved soil samples were then placed in 10 mL EP tubes, labeled, and stored in a liquid nitrogen tank for subsequent high-throughput sequencing of soil microorganisms.

### 2.3. Experimental Design

#### 2.3.1. Sequencing

Genomic DNA was extracted from soil samples using the CTAB method, and its concentration and purity were assessed via 1% agarose gel electrophoresis. The DNA was then diluted to 1 ng/μL for further analysis. PCR amplification targeted the bacterial 16S rRNA gene (V4–V5 regions) using specific primers (515F–907R). Each 30 μL reaction contained 15 μL Phusion^®^ High-Fidelity PCR Master Mix, 0.2 μM primers, and ~10 ng template DNA. The cycling conditions included initial denaturation (98 °C, 1 min), followed by 30 cycles of denaturation (98 °C, 10 s), annealing (50 °C, 30 s), and extension (72 °C, 60 s), with a final extension at 72 °C for 5 min. Amplifications were performed in triplicate, pooled, and verified via 2% agarose gel electrophoresis. The target bands were purified using an AxyPrep DNA Gel Extraction Kit (Axygen Scientific, Union City, CA, USA).

PCR products were quantified using a QuantiFluor™-ST Blue Fluorescence Quantification System (Promega, Madison, WI, USA) and pooled as per the sequencing requirements. For Illumina PE250 sequencing, library preparation involved adapter ligation, magnetic bead-based selection, PCR enrichment, and alkali denaturation to generate single-stranded DNA. Sequencing was performed by hybridizing DNA fragments to primers on a sequencing chip, forming bridge structures for cluster amplification. Fluorescently labeled dNTPs were incorporated, and the emitted signals were analyzed to determine nucleotide sequences.

#### 2.3.2. Bioinformatics Analysis Workflow

The Illumina PE250 sequencing reads were assembled based on their overlaps, followed by quality control and filtering. OTU clustering and taxonomic classification were performed. The diversity indices were calculated from OTU clustering results, and the sequencing depth was assessed. Taxonomic information was used for statistical analysis of community structure at various taxonomic levels. In-depth statistical and visualization analyses, such as community structure and phylogenetic analysis, were then conducted.

#### 2.3.3. Data Analysis

Raw sequencing data were processed using Trimmomatic for quality control, removing low-quality and adapter sequences. Paired-end reads were assembled with FLASH, retaining only those with overlapping regions. OTU clustering and taxonomic annotation were performed using UPARSE, grouping sequences with ≥97% similarity into the same OTU. Representative sequences were annotated with the RDP classifier based on the Greengenes or SILVA databases. Alpha diversity was assessed using the Chao1 index, the observed species, and the Shannon index, with rarefaction curves generated to evaluate the sequencing depth.

Beta diversity was analyzed through weighted and unweighted UniFrac distances, and bacterial community distributions were visualized with Krona plots. Hierarchical clustering based on the UniFrac distance was conducted using the UPGMA method. Differences in microbial taxonomy abundances were tested using Metastats, with multiple hypothesis correction via the Benjamini–Hochberg method. LEfSe analysis (LDA score ≥ 2.0) identified significant biomarkers, while ANOSIM and MRPP tested community differences based on the Bray–Curtis dissimilarity. Data visualization was performed with R or Origin 9.1, statistical analysis with SPSS Statistics 17.0 (*p* < 0.05), and image processing with Photoshop CS5.

## 3. Results and Analysis

### 3.1. Bacterial Diversity Analysis of Soil in the Estuarine Wetlands and Nearshore Wetlands of Qinghai Lake

#### 3.1.1. Sample Diversity Assessment (Alpha Diversity)

This study analyzed the species abundance and evenness of the samples using rank abundance curves. The results (Figure 2a,e) showed that species abundance at the 0–10 cm depth (a) was relatively high but differed significantly from that at the 10–20 cm depth (e), indicating depth-related variations in community structure. Rarefaction curves were used to assess the adequacy of sequencing depth and to compare the species richness of different samples. As shown in Figure 2b,f, the curves for 0–10 cm (b) and 10–20 cm (f) depths gradually plateaued with increasing sequencing depth, indicating that the sequencing depth was sufficient, and further increases contributed little to the species count. To evaluate microbial diversity, the Shannon–Wiener index was analyzed. As shown in Figure 2c,g, the Shannon index stabilized at different values for the 0–10 cm (c) and 10–20 cm (g) depths, reflecting distinct diversity patterns. Additionally, species accumulation curves (Figure 2d,h) for the 0–10 cm (d) and 10–20 cm (h) depths plateaued when the sample size approached 20, suggesting that the current sample size adequately captured the species richness at both depths.

#### 3.1.2. Alpha Diversity Parameter Test Analysis

In this study, parameter test analyses of bacterial alpha diversity were performed for different estuarine wetlands and nearshore estuarine wetlands at the 0–10 cm and 10–20 cm soil depths (Figure 3). The results showed that at the 0–10 cm soil depth, the ACE and Chao1 indices of the SC1a group were significantly lower than those of other groups (HC1a, DC1a, SC2a, HC2a) (*p* < 0.05), while the ACE and Chao1 indices of the HC1a group were significantly higher than those of the DC1a and SC2a groups (*p* < 0.05). No significant differences were observed between the other groups. Additionally, the Shannon index of the SC1a group was significantly lower than that of the HC1a, SC2a, and HC2a groups (*p* < 0.05), indicating that the microbial diversity in the SC1a group was lower. The Simpson index analysis revealed that the SC1a group had a significantly higher value compared to HC1a (*p* < 0.05), which may be related to the lower evenness of this group’s microbial community. At the 10–20 cm soil depth, the ACE and Chao1 indices of the HC1b group were significantly higher than those of the SC1b, DC1b, and SC2b groups (*p* < 0.05), while no significant differences were found between the other groups, indicating that the HC1b group had a higher species richness. The Shannon index did not show significant differences (*p* > 0.05), suggesting that the microbial diversity was similar between the groups. Regarding the Simpson index, the SC1b group had significantly lower values than the HC1b and DC2b groups (*p* < 0.05), indicating higher microbial community evenness in the SC1b group. Overall, there were considerable differences in microbial diversity across different wetlands and soil depths.

### 3.2. OTU Distribution Characteristics of Soil Bacteria in the Estuarine and Nearshore Wetlands of Qinghai Lake

This study systematically explored the distribution characteristics of bacterial operational taxonomic units (OTUs) across different wetland types and soil depths through Venn analysis and petal charts. The results revealed significant differences in the total number of OTUs and the number of unique OTUs among the sample groups, indicating that wetland type and soil depth have a significant impact on bacterial community structure.

In the 0–10 cm soil layer (Figure 4a), the total number of OTUs across all groups was 56,411, with the following OTU counts for each group: SC1a (6707), HC1a (10,765), DC1a (9825), SC2a (9646), HC2a (8461), and DC2a (11,007), which accounted for 11.89%, 19.08%, 17.42%, 17.10%, 15.00%, and 19.51% of the total OTUs, respectively. Among these, SC1a had the lowest proportion, while DC2a had the highest. The number of shared OTUs across all groups was 357, representing only 0.63% of the total OTUs. Furthermore, the number of unique OTUs for each group was as follows: SC1a (657), HC1a (1714), DC1a (1956), SC2a (1961), HC2a (885), and DC2a (2401), which accounted for 9.80%, 15.92%, 19.91%, 20.33%, 10.46%, and 21.81% of the total OTUs in each group, respectively. These results indicate that DC1a, SC2a, and DC2a exhibit a higher species uniqueness.

In the 10–20 cm soil layer (Figure 4b), the total number of OTUs across all groups was 47,041, with the following OTU counts for each group: SC1b (6835), HC1b (8637), DC1b (7672), SC2b (7503), HC2b (8027), and DC2b (8367), which accounted for 14.53%, 18.36%, 16.31%, 15.95%, 17.06%, and 17.79% of the total OTUs, respectively. Among these, SC1b had the lowest proportion, while HC1b had the highest. The number of shared OTUs across all groups was 229, representing 0.49% of the total OTUs. Additionally, the number of unique OTUs for each group was as follows: SC1b (983), HC1b (1540), DC1b (1291), SC2b (1305), HC2b (1170), and DC2b (2036), which accounted for 14.38%, 17.83%, 16.83%, 17.39%, 14.58%, and 24.33% of the total OTUs in each group, respectively. These results suggest that DC2b exhibits a higher species uniqueness.

### 3.3. Analysis of the Soil Bacterial Community Composition in the Estuarine and Nearshore Estuarine Wetlands of Qinghai Lake

#### 3.3.1. Phylum-Level Community Composition Analysis

Based on the 16S rRNA gene sequencing data, this study analyzed the bacterial community structure in the 0–10 cm soil layer of the estuarine and nearshore estuarine wetlands of Qinghai Lake (SC1a, SC2a, HC1a, HC2a, DC1a, DC2a), revealing significant diversity and spatial variation at the phylum level. A total of 74 bacterial phyla were detected according to the sequencing data, community composition plot (Figure 5a), and heatmap (Figure 5c). The predominant phyla included Proteobacteria, Actinobacteriota, Bacteroidota, Acidobacteriota, and Chloroflexi, which were dominant across all sampling sites. Additionally, lower-abundance phyla such as Cyanobacteria, Myxococcota, and Verrucomicrobiota were also detected. The abundance of Proteobacteria ranged from 35.09% (DC2a) to 52.71% (SC2a), showing a trend of higher abundance in the estuarine wetlands (SC2a, HC2a, DC2a) compared to the nearshore wetlands (SC1a, HC1a, DC1a). Actinobacteriota exhibited spatial heterogeneity, with the highest abundance in DC1a and relatively lower abundance in HC2a and DC2a. Bacteroidota had a significantly higher abundance in nearshore wetlands, particularly reaching its peak in HC1a. Among the secondary phyla, Chloroflexi and Acidobacteriota showed a higher abundance in estuarine wetlands compared to nearshore wetlands, indicating their potential stronger adaptation to the wetland’s water–salt environment. Furthermore, Myxococcota exhibited a higher abundance in HC2a and DC2a, suggesting that it may play a specific ecological role in estuarine wetland soils. Among the less common phyla, Cyanobacteria had the highest abundance in the nearshore wetland DC1a, while Verrucomicrobiota significantly increased in the estuarine wetland DC2a.

In the 10–20 cm soil layer, the bacterial community composition at different sampling sites (SC1b, SC2b, HC1b, HC2b, DC1b, and DC2b) exhibited significant spatial heterogeneity at the phylum level. Based on sequencing data, community composition plots (Figure 5b), and heatmap analysis (Figure 5d), a total of 67 bacterial phyla were detected. Among these, Proteobacteria, Actinobacteriota, Acidobacteriota, and Bacteroidota were the dominant phyla, which were prevalent across all sampling sites. Proteobacteria showed a generally high relative abundance across all sites but exhibited significant distribution differences. For instance, the relative abundance was highest in DC1b, while HC2b and DC2b exhibited a relatively lower abundance. Actinobacteriota, second in abundance to Proteobacteria, was significantly more abundant in HC2b compared to the other sites, while its abundance was lower in DC2b, indicating its sensitivity to environmental conditions. Acidobacteriota showed a higher abundance in SC1b and SC2b, but a significantly lower abundance in DC1b and DC2b. Bacteroidota had the highest abundance in DC1b, followed by SC2b and HC2b. Chloroflexi exhibited a higher abundance in HC2b and DC2b, with lower levels in SC1b and HC1b, indicating a gradient distribution. Patescibacteria had a higher relative abundance in SC1b and SC2b compared to HC2b and DC2b, showing a clear spatial distribution pattern. Myxococcota reached its peak relative abundance in DC2b, while it was lower in SC1b and HC1b. Firmicutes was most abundant in HC1b, with a lower abundance in SC2b and DC2b. Additionally, Nitrospirota had a significantly higher abundance in DC2b compared to the other sites, while SC1b and HC1b showed a lower abundance. Desulfobacterota was most abundant in DC2b. Verrucomicrobiota had a higher abundance in DC2b, while it was lower in SC1b and HC1b, showing some spatial differentiation. Gemmatimonadota reached its highest abundance in SC2b and its lowest in DC1b. Fusobacteriota was most abundant in HC1b, while it was least abundant in DC2b. Campylobacterota was most abundant in HC2b, but least abundant in SC2b. Finally, Calditrichota exhibited a significantly higher abundance in DC2b compared to other sampling sites.

#### 3.3.2. Family-Level Community Composition Analysis

Based on the 16S rRNA gene sequencing data, this study analyzed the bacterial community structure in the 0–10 cm soil layer of the Qinghai Lake estuarine wetlands and nearshore estuarine wetlands (SC1a, SC2a, HC1a, HC2a, DC1a, DC2a) (Figure 6a,c), revealing significant diversity and spatial heterogeneity at the family level. The major bacterial families include Sphingomonadaceae, Flavobacteriaceae, and Comamonadaceae, whose abundances varied significantly across different wetlands. For instance, Sphingomonadaceae exhibited the highest abundance in HC1a and the lowest in DC2a. Gemmatimonadaceae and Hydrogenophilaceae also displayed significant abundance changes, with Hydrogenophilaceae being most abundant in SC1a and least abundant in SC2a. Nitrosomonadaceae and Xanthomonadaceae showed relatively balanced abundances between the wetlands, but were significantly higher in HC1a. Vicinamibacterales_uncultured and Chitinophagaceae were more abundant in SC2a and HC1a, whereas Bacteroidetes_vadinHA17 was notably dominant in DC2a. Saprospiraceae and Subgroup_7_norank exhibited higher abundances in SC2a and HC2a, while Pyrinomonadaceae was more abundant in SC1a but nearly absent in DC2a. Caulobacteraceae and Anaerolineaceae showed significantly higher abundances in HC1a and DC2a, while Aeromonadaceae reached its highest abundance in SC2a, and it was relatively lower in SC1a and DC2a. Steroidobacteraceae and Rhizobiales_Incertae_Sedis were more abundant in HC2a and SC1a, with Pseudomonadaceae being most abundant in SC2a. Prolixibacteraceae and Oxalobacteraceae exhibited higher abundances in SC1a and HC2a, while Ilumatobacteraceae and Vicinamibacteraceae were more abundant in DC1a and SC2a. Sulfurimonadaceae and Geobacteraceae showed significant abundances in HC2a and DC2a, while Sutterellaceae and Thermoanaerobaculaceae were more abundant in DC2a and least abundant in SC1a and SC2a. Xanthobacteraceae and Desulfosarcinaceae were significantly abundant in HC2a and DC2a, respectively, while A4b and Actinomarinales_uncultured were more abundant in SC1a and DC1a. Micrococcaceae and Rhodocyclaceae were notably abundant in HC1a and SC2a, and S0134_terrestrial_group_norank was most abundant in DC1a. Thermodesulfovibrionia_uncultured and Crocinitomicaceae showed higher abundances in DC2a and HC2a, respectively, while Erysipelotrichaceae and Ignavibacteriaceae were most abundant in HC1a and DC2a. Additionally, SBR1031_norank and SC-I-84 exhibited higher abundances in DC2a, while PHOS-HE36 and Sphingobacteriaceae were significantly more abundant in SC1a and HC2a.

Based on the 16S rRNA gene sequencing data, this study analyzed the bacterial community composition in the 10–20 cm soil layer of the Qinghai Lake estuarine wetlands and nearshore estuarine wetlands (SC1b, HC1b, DC1b, SC2b, HC2b, DC2b) (Figure 6b,d). In these sites, Gemmatimonadaceae and Comamonadaceae were commonly present, with relatively high abundance and even distribution. Nitrosomonadaceae and Sphingomonadaceae also exhibited high abundances. Similarly, Pyrinomonadaceae and Vicinamibacterales_uncultured showed higher abundances across multiple sites, particularly in SC1b and SC2b. In addition, Anaerolineaceae and Chitinophagaceae were relatively abundant. Arcobacteraceae had the highest abundance in HC2b, while Flavobacteriaceae showed a distinct abundance peak in DC1b. Likewise, Sulfurimonadaceae and Hydrogenophilaceae had higher abundances in the HC2b and DC2b sites. Other important families included Bacteroidetes_vadinHA17, Xanthomonadaceae, Steroidobacteraceae, Geobacteraceae, and Desulfatiglandaceae. Rare families such as Paenibacillaceae, Shewanellaceae, and Ignavibacteriaceae had lower abundances. This study also found that Alphaproteobacteria_uncultured and Ilumatobacteraceae displayed relatively stable abundances across different sites.

#### 3.3.3. Genus-Level Structural Composition Analysis

Based on the 16S rRNA gene sequencing data, this study conducted a genus-level analysis of the bacterial community structure in the 0–10 cm soil layer of the Qinghai Lake estuarine and nearshore estuarine wetlands (SC1a, HC1a, DC1a, SC2a, HC2a, DC2a) (Figure 7a,c). The results revealed significant diversity and spatial heterogeneity of bacterial communities. At the genus level, *Thiobacillus* was significantly distributed in the SC1a and DC1a sites, but occurred at lower levels in other sites. *Sphingomonas* was relatively abundant in HC1a and HC2a but significantly reduced in DC2a. *Flavobacterium* dominated at the DC1a site, while being more evenly distributed at other sites. *Vicinamibacterales_uncultured* was notably abundant at the SC2a site. *Gemmatimonadaceae_uncultured* displayed similar distribution patterns at the SC2a and HC2a sites. Additionally, *Bacteroidetes_vadinHA17_norank* was significantly more abundant at the DC2a site compared to the other sites. *RB41* was more prevalent at the SC1a and SC2a sites but significantly reduced at HC1a and DC2a. *Aeromonas* was relatively dominant at the SC2a and HC2a sites. *Lysobacter* was relatively abundant at the SC1a and DC1a sites, but significantly decreased at sites in the nearshore estuarine wetlands. *Steroidobacteraceae_uncultured* was more abundant at the DC2a site. *Hydrogenophaga* was commonly distributed at SC1a and HC2a. *Ellin6067* was significantly enriched at the HC1a site, with a more even distribution at other sites. *Subgroup_7_norank* was most abundant at the HC2a site, but less so at SC1a. *Sulfuricurvum* showed a higher distribution at the DC2a site. *Sutterellaceae_uncultured* was relatively abundant at the HC2a site, with more uniform distribution at the other sites. *Anaerolineaceae_uncultured* was more abundant at the DC2a site. Other notable genera included *Actinomarinales_uncultured* at the DC1a site, *Ellin6055* at the SC2a site, *Fluviicola* at the HC2a site, and *Ignavibacterium*, which was notably distributed at the DC2a site.

The analysis of the bacterial community composition at the genus level in the 10–20 cm soil layer of the Qinghai Lake estuarine wetlands and nearshore estuarine wetlands (SC1b, SC2b, HC1b, HC2b, DC1b, DC2b) is shown in Figure 7b,d. The main dominant genera, such as *Vicinamibacterales_uncultured* and *Vicinamibacteraceae_norank*, exhibited a higher distribution at the SC1b and SC2b sites. *Gemmatimonadaceae_uncultured* showed a higher distribution at the SC2b site, and *RB41* had a higher abundance at the SC1b and SC2b sites. Some high-abundance genera with significant ecological functions include *Flavobacterium*, which was more abundant at HC1b and DC1b; *Pseudomonas*, which was more abundant at SC1b and SC2b; and *Pseudarcobacter*, which was more abundant at HC2b. Other functional bacteria, such as *Actinomarinales_uncultured*, showed a relatively stable distribution across all sites; *Sphingomonas* was more abundant at SC1b and HC1b; and *Sulfuricurvum* was concentrated at DC2b. Additionally, some low-abundance but ecologically important genera exhibited unique distributions at specific sites, such as *Aeromonas* at HC2b, *Thiobacillus* at DC2b, and *Trichococcus* at HC1b and DC1b. Characteristic genera, such as *Ellin6067*, were more abundant at DC1b, while *Rheinheimera* and *Brevundimonas* were more abundant at DC1b. Low-abundance genera such as *Nitrospira* and *Aminicenantales_norank* were also detected.

### 3.4. Weighted UniFrac Distance Analysis of Bacterial Communities in the Qinghai Lake Estuarine Wetlands and Nearshore Wetlands

As shown in Figure 8a, the overall distribution of weighted UniFrac distances for the “All within Description” group is relatively small, indicating high similarity and consistency among the samples within the group. In contrast, the “All between Description” group has a wider distance distribution, reflecting significant differences in characteristics between the samples across the groups. In the within-group comparisons, the distance distribution between HC1a vs. HC1a is the lowest, showing that the samples within this group have the most consistent characteristics. The distance distribution for DC1a vs. DC1a is the highest, indicating relatively large differences in characteristics between the samples within this group. In the between-group comparisons, the distance distribution between HC1a and HC2a is the lowest, suggesting that these two groups of samples have high similarity, likely due to similar environmental or ecological conditions. In contrast, the distance distributions for HC2a vs. SC2a, HC2a vs. SC1a, SC1a vs. HC1a, and SC1a vs. DC1a are smaller and narrower, indicating that the characteristics of these cross-group samples are relatively close, with higher internal consistency. The distance distribution between SC2a and DC2a has the widest range, suggesting significant differences in characteristics between these two groups.

As shown in Figure 8b, the overall trend indicates that the “All within Description” group has the smallest range of weighted UniFrac distances, suggesting high similarity and small differences between the samples within the group. On the other hand, the “All between Description” group has the largest range of distances, reflecting significant differences in characteristics between the samples across the groups. In the within-group comparisons, the distance distribution between SC1b and SC1b is the lowest, indicating high consistency within the group. The distance distribution for DC1b vs. DC1b is the highest, showing relatively large differences in characteristics between the samples within this group. In the between-group comparisons, the distance distributions between SC1b and HC2b and between SC1b and SC2b are the lowest, indicating that these two groups of samples have similar characteristics, possibly due to similar environmental or ecological conditions. Furthermore, the distance distributions for HC1a vs. HC2b, HC2b vs. SC2b, and HC1b vs. SC2b are smaller and narrower, suggesting that the characteristics of these pairs of samples are relatively close, with strong internal consistency. The distance distribution between SC2b and DC2b has the largest range, indicating the most significant differences in characteristics between these two groups.

### 3.5. LEfSe LDA of Soil Bacterial Communities in the Estuarine and Nearshore Wetlands of Qinghai Lake

LEfSe analysis was used to detect significant differences in bacterial abundances across the soil samples and identify taxa with statistical significance between the groups. Combined with linear discriminant analysis (LDA), this method evaluated the contribution of each species to intergroup differences.

In the 0–10 cm soil layer (Figure 9a), the numbers of significantly different bacterial taxa identified (*p* < 0.05) were as follows: 621 for HC1a, 562 for SC1a, 602 for DC1a, 586 for HC2a, 563 for SC2a, and 596 for DC2a. In the HC1a group, 69 genera showed highly significant differences (*p* < 0.01), including *Brevifollis*, *Candidatus Methylomirabilis*, *Candidatus Saccharimonas*, *Jeotgalicoccus*, *Cnuella*, *Arcicella*, *Defluviimonas*, *Fluviicoccus*, *Vitellibacter*, *Pajaroellobacter*, *Rubellimicrobium*, *P3OB-42*, *Rubrobacter*, *Microvirga*, *Syntrophorhabdus*, *Microlunatus*, *Flavisolibacter*, *Desulfovirga*, *Thiobacillus*, *Sediminibacterium*, *Rhodocytophaga*, *FFCH7168*, *Gemmatimonadaceae*, *Acidibacter*, and *Spirochaeta*. These genera were distributed across multiple phyla, including Proteobacteria, Firmicutes, Cyanobacteria, Bacteroidota, Myxococcota, Actinobacteriota, Spirochaetota, Desulfobacterota, Chloroflexi, and Gemmatimonadota. In the SC1a group, 43 genera exhibited highly significant differences (*p* < 0.01), such as *Thalassospira*, *Pajaroellobacter*, *Rubellimicrobium*, *P3OB-42*, *Rubrobacter*, *Sediminispirochaeta*, *Microvirga*, *Syntrophorhabdus*, *Microlunatus*, *Flavisolibacter*, *Desulfovirga*, *Thiobacillus*, *FFCH7168*, *SM23-31*, *Acidibacter*, and *Spirochaeta*. These taxa spanned phyla such as Proteobacteria, Firmicutes, Bacteroidota, Desulfobacterota, Spirochaetota, and Chloroflexi. In the DC1a group, 41 genera demonstrated highly significant differences (*p* < 0.01), including *Leptolyngbya RV74*, *Caproiciproducens*, *Chitinispirillum*, *Thalassospira*, *Vitellibacter*, *Pajaroellobacter*, *Rubellimicrobium*, *P3OB-42*, *Rubrobacter*, *Sediminispirochaeta*, *Flavisolibacter*, *Desulfovirga*, *Thiobacillus*, *FFCH7168*, *Acidibacter*, *Methylocaldum*, and *Spirochaeta*. These taxa were associated with phyla such as Proteobacteria, Firmicutes, Cyanobacteria, Bacteroidota, Myxococcota, Actinobacteriota, Spirochaetota, Desulfobacterota, and Chloroflexi. In the HC2a group, 35 genera showed highly significant differences (*p* < 0.01), including *Defluviimonas*, *Fluviicoccus*, *Pajaroellobacter*, *Rubellimicrobium*, *P3OB-42*, *Rubrobacter*, *Sediminibacterium*, *Rhodocytophaga*, *Syntrophorhabdus*, *Microlunatus*, *Flavisolibacter*, *Desulfovirga*, *Thiobacillus*, *FFCH7168*, *SM23-31*, *Acidibacter*, and *Spirochaeta*. These genera were distributed across phyla such as Proteobacteria, Firmicutes, Bacteroidota, Desulfobacterota, Spirochaetota, Chloroflexi, and Acidobacteriota. In the SC2a group, 38 genera were highly significantly different (*p* < 0.01), such as *Ekhidna*, *Rhabdobacter*, *Nostoc 0708*, *NS10 marine group*, *Pajaroellobacter*, *Rubellimicrobium, P3OB-42*, *Rubrobacter*, *Flavisolibacter, Desulfovirga*, *Thiobacillus*, *FFCH7168*, *Acidibacter*, and *Spirochaeta*. These taxa belonged to phyla such as Proteobacteria, Firmicutes, Cyanobacteria, Bacteroidota, Myxococcota, Spirochaetota, and Actinobacteriota. In the DC2a group, 43 genera exhibited highly significant differences (*p* < 0.01), including *Maritimimonas*, *Prevotella*, *Latescibacter*, *Elusimicrobiota*, *Coriobacteriia*, *Colwellbacteria*, *Cloacimonadia*, *Methylomagnum*, *Azospirillales*, *Anaerolineae*, *Ignavibacteria*, *Dehalococcoidia*, *Pajaroellobacter*, *P3OB-42*, *Rubrobacter*, *Sediminispirochaeta*, *Chitinophagaceae*, *Microvirga*, *Gallionellaceae*, *Acidobacteriae*, *Fermentibacteraceae*, *Christensenellaceae*, *Syntrophorhabdus*, *Flavisolibacter*, *Desulfovirga*, *Kryptonia*, *Thiobacillus*, *Sediminibacterium*, *FFCH7168*, *Pleomorphomonadaceae*, *SM23-31*, *Gemmatimonadaceae*, *Methylocaldum*, *Acidibacter*, *Spirochaeta*, and *Frankiales*. These taxa covered a wide range of phyla, including Bacteroidota, Proteobacteria, Chloroflexi, Spirochaetota, and Actinobacteriota.

Through sequencing analysis, the significant differential bacterial taxa (*p* < 0.05) identified in the 10–20 cm soil layer (Figure 9b) across the groups were as follows: HC1b group (504 taxa), SC1b group (453 taxa), DC1b group (490 taxa), HC2b group (481 taxa), SC2b group (450 taxa), and DC2b group (488 taxa). The HC1b group exhibited 41 bacterial genera that showed highly significant differences (*p* < 0.01), including *Cnuella*, *Niabella*, *Candidatus Chlorothrix*, *Acetobacteroides*, *Jeotgalicoccus*, *Ercella*, *UKL13-1*, *Candidatus Contendobacter*, *Rhodocytophaga*, *Polymorphobacter*, *Paracoccus*, *Lacihabitans*, *Flavitalea*, *Roseiflexus*, *Dinghuibacter*, *P3OB-42*, *Desulfococcus*, *Lacibacter*, *Gillisia*, *YC-ZSS-LKJ147*, and *Pseudoduganella*. These genera belong to phyla such as Bacteroidota, Chloroflexi, Firmicutes, Proteobacteria, Actinobacteriota, Desulfobacterota, Methylomirabilota, Gemmatimonadota, Myxococcota, and Patescibacteria. In the SC1b group, 27 bacterial genera, including *Parvibaculum*, *Paracoccus*, *Thermomonas*, *Sphingorhabdus*, *Lacihabitans*, *Flavitalea*, *Lacibacter*, *Acidovorax*, *Gillisia*, and *P3OB-42*, demonstrated highly significant differences (*p* < 0.01). These genera were distributed across phyla such as Actinobacteriota, Proteobacteria, Entotheonellaeota, Chloroflexi, Bacteroidota, Desulfobacterota, Methylomirabilota, Gemmatimonadota, Myxococcota, and Patescibacteria. In the DC1b group, 35 bacterial genera showed highly significant differences (*p* < 0.01), including *Incertae Sedis*, *Marinifilum*, *Planktosalinus*, *Alkaliphilus*, *Marinifilaceae*, *Paracoccus*, *Thermomonas*, *Sphingorhabdus*, *Flavitalea*, *Lacibacter*, *Acidovorax*, *Gillisia*, *P3OB-42*, *Desulfococcus*, *Lacihabitans*, *TPD-58*, and *YC-ZSS-LKJ147*. These genera belong to phyla such as Firmicutes, Patescibacteria, Proteobacteria, Bacteroidota, Fermentibacterota, Latescibacterota, Myxococcota, Chloroflexi, Entotheonellaeota, Acidobacteriota, and Gemmatimonadota. In the HC2b group, 28 bacterial genera demonstrated highly significant differences (*p* < 0.01), including *Paracoccus*, *Thermomonas*, *Sphingorhabdus*, *Lacihabitans*, *Flavitalea*, *Acidovorax*, *Roseiflexus*, *Dinghuibacter*, *P3OB-42*, *TPD-58*, *Gillisia*, and *Pseudoduganella*. These genera were distributed across phyla such as Proteobacteria, Chloroflexi, Entotheonellaeota, Actinobacteriota, Bacteroidota, Desulfobacterota, Methylomirabilota, Gemmatimonadota, Myxococcota, Patescibacteria, and Acidobacteriota. In the SC2b group, 26 bacterial genera, including *Rhodocytophaga*, *Paracoccus*, *Entotheonellaeota*, *Thermomonas*, *Sphingorhabdus*, *Lacihabitans*, *Flavitalea*, *Acidovorax*, *Roseiflexus*, *Dinghuibacter*, *P3OB-42*, *YC-ZSS-LKJ147*, and *Pseudoduganella*, showed highly significant differences (*p* < 0.01). These genera belong to phyla such as Bacteroidota, Proteobacteria, Entotheonellaeota, Chloroflexi, Actinobacteriota, Desulfobacterota, Methylomirabilota, Gemmatimonadota, Myxococcota, and Patescibacteria. In the DC2b group, 41 bacterial genera showed highly significant differences (*p* < 0.01), including *Cutibacterium*, *Rickettsiella*, *Actibacter*, *Desulforhabdus*, *Z114MB74*, *Polymorphobacter*, *Paracoccus*, *Thermomonas*, *Sphingorhabdus*, *Lacihabitans*, *Desulfococcus*, *Lacibacter*, *Dinghuibacter*, *P3OB-42*, *TPD-58*, and *Gillisia*. These genera were distributed across phyla such as Actinobacteriota, Armatimonadota, Cloacimonadota, Proteobacteria, Acidobacteriota, Bacteroidota, Chloroflexi, Fibrobacterota, Methylomirabilota, Fermentibacterota, Latescibacterota, Myxococcota, Gemmatimonadota, and Patescibacteria.

## 4. Discussion

### 4.1. Comparative Analysis of Soil Bacterial Diversity Between the Estuarine and Nearshore Estuarine Wetlands of Qinghai Lake

This study analyzed the alpha diversity and OTU distribution characteristics of soil bacterial communities in the estuarine and nearshore estuarine wetlands of Qinghai Lake, exploring the differences between these two wetland types and their potential ecological drivers. Alpha diversity analysis indicated that wetland type and soil depth significantly influenced microbial diversity. In the 0–10 cm soil layer, the ACE and Chao1 indices of group SC1a were significantly lower than those of other groups, suggesting reduced species richness. In contrast, group HC1a exhibited the highest species richness, with significantly higher ACE and Chao1 indices than groups DC1a and SC2a. The Shannon and Simpson indices further supported this conclusion, showing that the Shannon index of SC1a was significantly lower than that of HC1a, SC2a, and HC2a, indicating reduced microbial diversity and community evenness in SC1a. These findings align with studies of saline wetland ecosystems, such as those by Zhang et al. [28], who reported lower bacterial richness in high-salinity estuarine soils due to osmotic stress, a condition likely prevalent in SC1a given its proximity to Qinghai Lake’s saline waters [29,30]. However, the high richness observed in HC1a contrasts with results from Li et al. [31] in a tidal freshwater wetland of the Yellow River Delta, where lower richness was attributed to limited nutrient availability. This discrepancy may reflect localized geochemical conditions or a stronger vegetation influence in HC1a, potentially enhancing microbial niches through root exudates. Previous studies suggest that plant root exudates can promote microbial diversity by providing labile carbon sources, which could contribute to the observed patterns, though further experimental validation is needed. In the 10–20 cm soil layer, the ACE and Chao1 indices of group HC1b were significantly higher than those of SC1b, DC1b, and SC2b, indicating increased species richness in HC1b. However, the Shannon index showed no significant differences, suggesting that microbial diversity remained relatively stable across the groups at this depth. The Simpson index indicated a higher evenness in SC1b, reflecting a more uniform bacterial community. These patterns partially align with the results of Wang et al. [32], who found that deeper soil layers in estuarine wetlands exhibit stable diversity due to reduced oxygen and nutrient gradients. However, the elevated richness in HC1b, compared to their findings, may be attributed to localized sediment deposition and long-term organic matter accumulation rather than direct nutrient input from upstream tributaries [33,34].

Analysis of OTU distribution further elucidated the impact of wetland type and soil depth on bacterial community structure. In the 0–10 cm soil layer, groups DC2a, SC2a, and DC1a exhibited a higher proportion of unique OTUs, reflecting distinct microbial communities across wetland types. Conversely, SC1a showed the lowest proportion of unique OTUs, indicating a less complex community structure, potentially constrained by salinity stress. This pattern aligns with observations in high-altitude saline wetlands, where increased salinity often reduces microbial diversity and phylogenetic uniqueness [35]. In the 10–20 cm soil layer, group DC2b exhibited the highest proportion of unique OTUs, while SC1b had the lowest, suggesting a more homogeneous microbial community at this depth. Previous studies in saline wetlands of the Yellow River Delta reported that deeper soil layers generally exhibit lower uniqueness due to substrate limitations [36]. However, the high OTU uniqueness observed in DC2b may indicate localized ecological heterogeneity, potentially driven by historical sediment deposition or long-term organic matter accumulation.

Overall, significant differences in bacterial diversity and OTU distribution exist between the estuarine and nearshore estuarine wetlands of Qinghai Lake, driven by wetland type, soil depth, and associated ecological factors. By comparing our results with studies of similar habitats, such as saline plateau wetlands and estuarine systems, this study highlights the role of salinity and vegetation as the key drivers of microbial diversity. Additionally, Qinghai Lake’s high-altitude saline conditions likely contribute to distinctive bacterial community patterns, though further comparative studies across different highland saline wetlands are needed to validate these findings. These insights enhance our understanding of microbial ecology in extreme wetland ecosystems and provide a foundation for future functional studies in this region.

### 4.2. Regional Variations in the Soil Bacterial Community Structure and Functional Potential Between the Estuarine Wetlands and Nearshore Estuarine Wetlands of Qinghai Lake

The soil bacterial communities in estuarine wetlands and nearshore estuarine wetlands of Qinghai Lake exhibit significant regional differences at the phylum, family, and genus levels, likely driven by water salinity gradients, soil physicochemical properties, and vegetation coverage [37,38,39]. Xia et al. (2020) reported that salinity and soil organic matter in estuarine wetlands exhibit a gradient distribution [40], a pattern consistent with our observations. This section explores the composition of major bacterial communities and their potential ecological functions, integrating comparisons with similar studies to elucidate these variations.

Proteobacteria was the dominant phylum across all sites, with a significantly higher relative abundance in the estuarine wetlands (SC2a, HC2a, DC2a) compared to the nearshore estuarine wetlands (SC1a, HC1a, DC1a). This variation may be attributed to environmental heterogeneity, particularly differences in moisture and nutrient availability, which selectively promote different Proteobacteria subgroups. For example, copiotrophic Gammaproteobacteria tend to dominate nutrient-rich environments, whereas Alphaproteobacteria are often associated with plant symbioses and oligotrophic conditions [41]. Actinobacteriota was more abundant in the nearshore estuarine wetlands, suggesting an adaptation to lower salinity and potentially higher organic matter inputs, consistent with findings from coastal wetlands [42]. Acidobacteriota exhibited a higher relative abundance in the estuarine wetlands, reflecting its tolerance to saline and moist conditions, which aligns with observations in acidic saline soils [43]. Secondary phyla showed distinct distribution patterns. Chloroflexi was more abundant in the estuarine wetlands, likely due to its role in organic matter decomposition under anaerobic or microaerophilic conditions. Myxococcota, which contributes to predatory interactions and complex organic matter degradation, was enriched in HC2a and DC2a, potentially influenced by substrate availability. The increased presence of Cyanobacteria in the nearshore estuarine wetlands suggests an adaptation to lower salinity and photic environments, where certain taxa can thrive under eutrophic conditions and contribute to nitrogen fixation, as previously observed in organic-rich wetland soils [44].

Environmental conditions drove distinct family-level differentiation. Sphingomonadaceae was most abundant in HC1a, potentially due to its association with low-salinity, vegetation-rich environments, a pattern observed in vegetated ecosystems [45]. The prevalence of Flavobacteriaceae in DC1a suggests a preference for environments with substantial organic input and moderate moisture levels [44]. In the estuarine wetlands, Sulfurimonadaceae and Hydrogenophilaceae were enriched, indicating active sulfur oxidation and denitrification processes [46,47]. The increased presence of Anaerolineaceae suggests anaerobic conditions favoring fermentative carbon cycling [48], a trend reported in anoxic wetland soils [43]. Notably, Desulfobacterota was enriched in the estuarine wetlands, highlighting the importance of sulfate reduction in the biogeochemical cycles of these ecosystems. Similarly, Geobacteraceae, known for its role in iron reduction, was more abundant in the estuarine wetland sites, supporting the role of redox-active minerals in carbon and nutrient dynamics [49,50].

At the genus level, microbial ecological adaptations were evident. The high relative abundance of Nitrosomonas in SC2a and HC2a suggests active ammonia oxidation, a key step in nitrogen cycling [51], consistent with previous reports in saline wetlands [41]. The enrichment of Pseudomonas in SC2a indicates a potential role in hydrocarbon degradation and organic matter turnover [52], which is characteristic of nutrient-rich soils [45]. Desulfovibrio and Shewanella, both enriched in HC2a and DC2a, contribute to sulfate and iron reduction, respectively [49,53], influencing carbon and nutrient cycling. The presence of Methanobacterium suggests localized methanogenic activity, likely occurring in anaerobic microsites with elevated hydrogen availability. As a hydrogenotrophic methanogen, Methanobacterium utilizes H₂ and CO₂ for methane production, a process relevant to carbon flux dynamics in high-altitude wetlands, particularly under conditions of high organic matter turnover and sulfate depletion [43,54].

The high salinity and nutrient complexity of the Qinghai Lake estuarine wetlands favor salt-tolerant taxa such as Acidobacteriota and Chloroflexi, while the lower salinity of the nearshore estuarine wetlands supports Actinobacteriota and Cyanobacteria, a pattern consistent with broader saline–freshwater wetland studies [41,44]. Vegetation influences these communities by shaping root exudate profiles, selectively promoting specific microbial taxa. Functionally, the estuarine wetlands are characterized by enhanced anaerobic decomposition and sulfate reduction, whereas the nearshore estuarine wetlands exhibit higher nitrogen fixation and aerobic organic matter degradation potential. These findings provide novel insights into the microbial ecology of high-altitude, saline wetland systems and have implications for wetland conservation strategies in Qinghai Lake.

### 4.3. Spatial Distribution and Ecological Characteristics of Soil Bacterial Communities in the Estuarine and Nearshore Estuarine Wetlands of Qinghai Lake

This study employed weighted UniFrac distance analysis and LEfSe LDA to examine the spatial distribution and phylogenetic divergence of soil bacterial communities in the estuarine and nearshore estuarine wetlands of Qinghai Lake. These analyses revealed significant compositional differences across wetland types, highlighting the ecological complexity and environmental drivers shaping microbial communities.

The weighted UniFrac analysis revealed that bacterial communities within the estuarine and nearshore estuarine wetlands exhibited relatively low phylogenetic distances, suggesting a high degree of evolutionary relatedness, likely driven by shared environmental conditions such as salinity and soil moisture [55]. In contrast, comparisons between the estuarine and nearshore estuarine wetlands (SC2a vs. DC2a, SC2b vs. DC2b) exhibited significantly greater phylogenetic distances, indicating pronounced divergence in microbial lineage composition.

Within individual wetland types, pairwise comparisons such as HC1a vs. HC1b and SC1a vs. SC1b showed the lowest distances, reflecting relative consistency in bacterial community composition within surface soils. This trend is likely influenced by uniform vegetation cover and organic matter input, consistent with previous findings in plateau wetlands [56]. Conversely, deeper soil layers (DC1a vs. DC1b) exhibited higher phylogenetic distances, suggesting increased microbial differentiation due to vertical stratification in oxygen availability and nutrient composition.

LEfSe analysis identified significant taxonomic and functional differences between the estuarine and nearshore estuarine wetlands. Proteobacteria was the most dominant phylum across all the wetland types, while Firmicutes exhibited higher relative abundance in the estuarine wetland sites, particularly in deeper layers, where anaerobic conditions prevail. These bacterial groups play crucial roles in nutrient cycling, including organic matter degradation and sulfur metabolism. In HC1a, Candidatus Methylomirabilis and Thiobacillus were significantly enriched, supporting active methane oxidation and sulfur oxidation, respectively [55]. In SC1a, Thalassospira and Pajaroellobacter were prominent, likely involved in phosphonic acid degradation and denitrification [57,58]. In deeper soils (DC1b), Desulfovirga and Flavisolibacter exhibited significant enrichment, supporting sulfate reduction and soil aggregation [49,59]. Notably, Spirochaeta showed increased abundance in the nearshore estuarine wetlands, suggesting adaptation to microaerophilic or fluctuating oxygen conditions, rather than strictly aerobic niches [60]. The enrichment of Acidibacter in the nearshore estuarine wetlands suggests its involvement in organic acid metabolism and soil pH regulation.

The spatial variation in bacterial communities between estuarine and nearshore estuarine wetlands is primarily driven by differences in soil moisture, salinity, and vegetation inputs. In surface soils, root exudates selectively enrich rhizosphere-associated taxa such as Sphingomonadaceae and Flavobacteriaceae, which participate in carbon and nitrogen cycling. In deeper soils, anaerobic conditions favor sulfate-reducing bacteria (Desulfovirga) and methanogenic archaea (Methanobacterium), driving key biogeochemical processes such as sulfate reduction and methane production. Vertical stratification in bacterial communities reflects a strong linkage between microbial metabolism and redox fluctuations. In oxygen-rich surface soils, heterotrophic respiration and nitrogen mineralization predominate, whereas in deeper, oxygen-depleted layers, anaerobic processes such as sulfate reduction (Desulfobacterota), iron reduction (Geobacteraceae), and methanogenesis (Methanobacterium) become more dominant. These depth-dependent microbial processes are critical for understanding carbon and nutrient cycling in wetland ecosystems.

### 4.4. Prediction of Functional Genes in Soil Bacterial Communities of the Estuarine Wetlands and Nearshore Estuarine Wetlands of Qinghai Lake

This study analyzed the KEGG metabolic pathway differences between the 0–10 cm and 10–20 cm soil layers of the estuarine and nearshore estuarine wetlands in Qinghai Lake, revealing major microbial metabolic pathways (Figure 10a,b) and their group-specific characteristics. Energy metabolism was the most dominant functional category across all groups, primarily involving carbohydrate catabolism and oxidative phosphorylation, which play crucial roles in organic matter decomposition and carbon cycling [61]. The abundance of genes related to energy metabolism was notably higher in DC2a, SC2a, DC1b, and DC2b, indicating active microbial involvement in carbon transformation. Carbohydrate metabolism exhibited significant variation, with HC1a and HC1b showing higher abundances of genes associated with carbohydrate utilization compared to HC2a and HC2b. This suggests potential differences in microbial contributions to soil organic matter turnover, a pattern previously observed in nutrient-rich ecosystems [62,63]. Amino acid metabolism was prevalent across multiple groups, indicating its role in microbial nitrogen processing, including protein turnover and biomass synthesis [64]. The high transcriptional activity observed in HC2a and HC2b suggests enhanced gene expression and metabolic adaptability [65]. Lipid metabolism was more pronounced in SC2a, DC1b, and DC2b, with SC2a exhibiting a particularly high abundance of genes related to fatty acid degradation. This suggests an increased microbial potential for lipid metabolism, a trait commonly associated with organic-rich or anoxic soil environments [66], as noted by Canfora et al. [41]. HC2a, HC1b, and HC2b exhibited high replication and repair pathway abundance, particularly in HC1b, suggesting active DNA maintenance and stress response mechanisms [67]. Xenobiotic degradation pathways were notably enriched in SC2a, DC2a, DC1b, and DC2b, indicating potential microbial involvement in the transformation of naturally occurring recalcitrant compounds, such as lignin derivatives or secondary metabolites from wetland vegetation [68]. Additionally, genes related to cell growth, death, and motility were more abundant in DC1a and SC2a, suggesting enhanced microbial adaptability in these environments [69]. SC1a and SC2a exhibited superior environmental adaptation, likely reflecting wetland-specific salinity and moisture gradients. Enzyme families displayed high activity, with SC1b showing notable catalytic capacity [70]. Protein metabolism was balanced across groups, with HC2a showing elevated gene expression related to protein synthesis and toxin resistance, a trait vital in saline habitats [58]. The biosynthesis of secondary metabolites was higher in surface layers (0–10 cm) than in deeper layers (10–20 cm), consistent with findings of Baumann et al. [71]. This was particularly pronounced in SC2a, HC2a, HC1a, and DC1a, suggesting that microbial communities in these regions contribute to ecological functions through the production of bioactive compounds.

Metabolic functions exhibited clear depth stratification. Genes associated with energy metabolism, carbohydrate metabolism, amino acid metabolism, and genetic information processing were universally abundant, underscoring their critical roles in carbon and nitrogen cycling [61,64]. Surface soils (0–10 cm), particularly in SC2a and DC2a, showed heightened gene abundance related to signal transduction, motility, and adaptation, supporting microbial contributions to organic matter decomposition [63]. In contrast, deeper soils (10–20 cm), especially in DC1b and DC2b, exhibited higher abundances of genes related to lipid metabolism, xenobiotic degradation, and DNA repair. This suggests potential microbial adaptations to anoxic conditions, where lipid metabolism and stress response mechanisms play a crucial role [72]. The enrichment of DNA repair pathways in deeper layers further suggests that microbial communities experience and mitigate genetic stress in these environments.

The microbial communities in the Qinghai Lake’s estuarine and nearshore estuarine wetlands exhibit unique functional adaptations to the high-altitude, saline environment. In surface soils, microbial communities prioritize adaptation and organic matter degradation, thereby enhancing carbon cycling. In deeper soils, microbial communities rely more on lipid metabolism and stress response mechanisms, potentially influencing nitrogen cycling. These findings highlight the distinct microbial functionalities in Qinghai Lake wetlands, improving our understanding of microbial ecosystem resilience and supporting conservation strategies for high-altitude saline wetlands.

## 5. Conclusions

This study provides the first in-depth characterization of bacterial community composition and function across soil depths in the estuarine and nearshore estuarine wetlands of Qinghai Lake, a high-altitude, hypersaline ecosystem. Unlike previous studies that report a uniform decline in microbial diversity with increasing salinity [28,31], our findings reveal distinct depth-dependent and habitat-specific responses. The nearshore wetlands exhibited a significantly lower alpha diversity in surface soils (0–10 cm; e.g., reduced ACE and Chao1 indices in SC1a) due to osmotic stress, whereas deeper soils (10–20 cm; e.g., HC1b) unexpectedly harbored high species richness, likely driven by sediment deposition and organic matter accumulation unique to this ecosystem. These results contrast with those from the Yellow River Delta and other coastal wetlands, where microbial richness tends to decline with depth due to nutrient limitation. Taxonomic and functional differentiation between the wetland types was also evident. The estuarine wetlands were dominated by Proteobacteria, Actinobacteriota, and Chloroflexi, with sulfate-reducing genera (*Desulfovibrio*, *Desulfobacter*) facilitating sulfur cycling; they were less prominent in the nearshore wetlands, where nitrogen-fixing Cyanobacteria (*Anabaena*, *Nostoc*) were enriched. This finding highlights an overlooked nitrogen fixation capacity in saline wetlands, which has not been well-documented in high-altitude ecosystems. Furthermore, KEGG functional predictions demonstrated that the surface soils (0–10 cm) in SC2a and DC2a were enriched in the pathways related to organic matter decomposition and signal transduction, whereas the deeper soils (10–20 cm) in DC1b and DC2b exhibited higher lipid metabolism and DNA repair, indicative of anaerobic adaptations. The higher OTU uniqueness in DC2b at 10–20 cm suggests localized ecological heterogeneity, possibly influenced by Qinghai Lake’s extreme salinity and altitude, distinguishing this wetland from previously studied lowland saline systems [36].

This study offers novel insights into how microbial communities mediate carbon, nitrogen, and sulfur cycling under high-altitude, hypersaline conditions, based on molecular sequencing and functional predictions. Unlike previous research that assumes salinity consistently constrains microbial diversity and function, our findings reveal habitat- and depth-specific adaptations that drive microbial heterogeneity. However, we acknowledge that these functional insights are inferred from genomic data rather than direct physiological measurements (e.g., rates of sulfate reduction or methane production), representing a limitation of this study. These results provide an ecological baseline for Qinghai Lake and emphasize the importance of considering microhabitat variation when formulating wetland conservation strategies. Future research should integrate physiological assays to quantify key processes such as sulfate reduction and nitrogen fixation, alongside exploring how soil properties (e.g., organic carbon, pH) and water–salt gradients shape microbial resilience in extreme wetland environments.

## Figures and Tables

**Figure 1 microorganisms-13-00759-f001:**
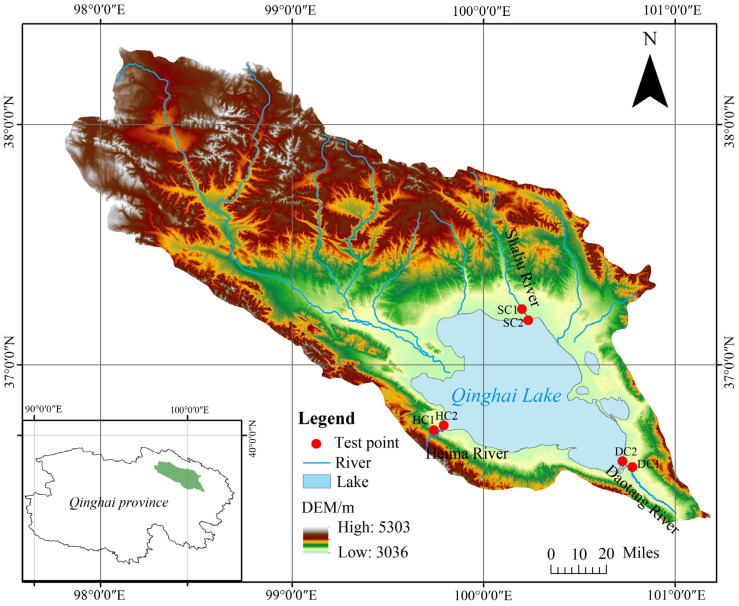
Location of the Qinghai Lake basin and our test plots.

**Figure 2 microorganisms-13-00759-f002:**
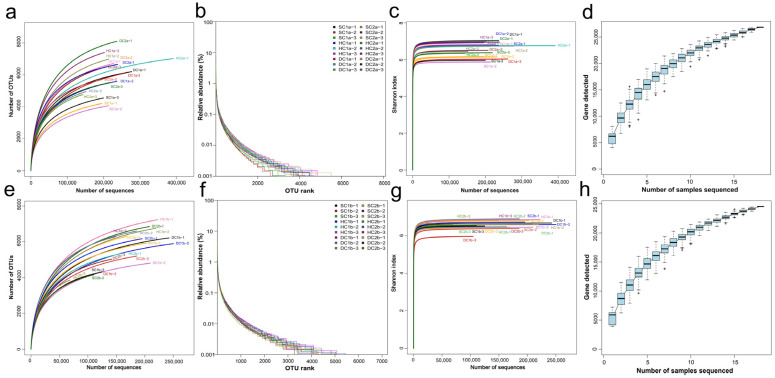
Sample diversity analysis. (**a**,**e**) Rank abundance curves for the soil depths of 0–10 cm and 10–20 cm, respectively; (**b**,**f**) rarefaction curves for the soil depths of 0–10 cm and 10–20 cm, respectively; (**c**,**g**) Shannon–Wiener curves for the soil depths of 0–10 cm and 10–20 cm, respectively; (**d**,**h**) species accumulation curves for the soil depths of 0–10 cm and 10–20 cm, respectively.

**Figure 3 microorganisms-13-00759-f003:**
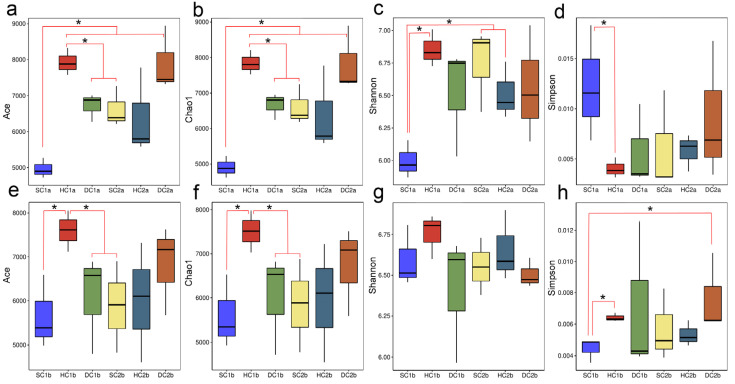
Box plot of alpha diversity indices across soil depths. (**a**,**e**) ACE indices for the 0–10 cm and 10–20 cm depths, respectively; (**b**,**f**) Chao1 indices for the 0–10 cm and 10–20 cm depths, respectively; (**c**,**g**) Shannon indices for the 0–10 cm and 10–20 cm depths, respectively; (**d**,**h**) Simpson indices for the 0–10 cm and 10–20 cm depths, respectively. The X-axis represents sample groups, and the Y-axis represents alpha diversity index values. Note: * indicates *p* < 0.05.

**Figure 4 microorganisms-13-00759-f004:**
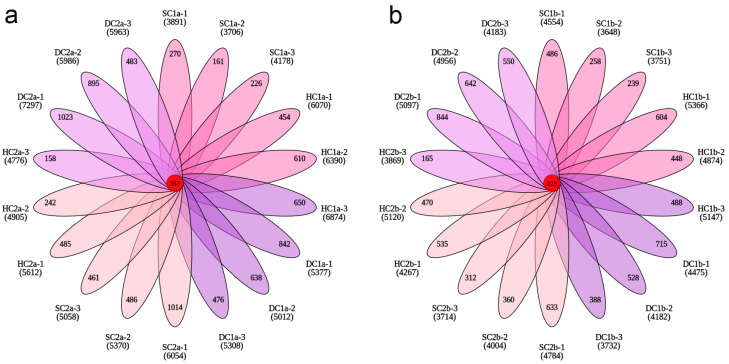
Petal plots of OTU numbers across soil depths. (**a**) OTU distribution in the 0–10 cm soil layer; (**b**) OTU distribution in the 10–20 cm soil layer. Each petal, distinguished by color, represents a unique sample. Overlapping regions indicate shared OTUs between the samples, with numbers labeled accordingly. The central number denotes OTUs common to all samples.

**Figure 5 microorganisms-13-00759-f005:**
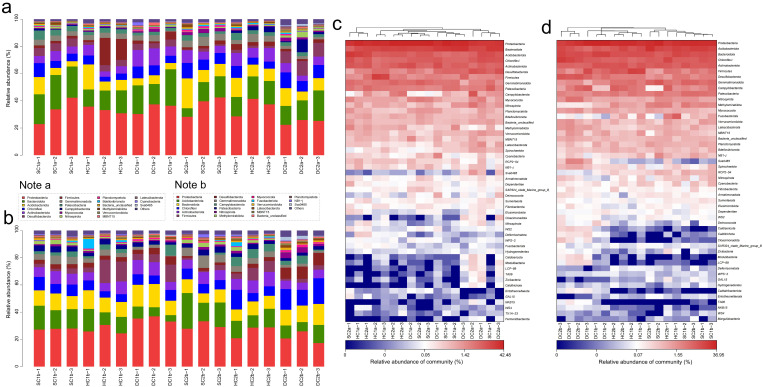
Community structure and composition analysis of soil bacteria at the phylum level. (**a**,**b**) Phylum-level bacterial community composition diagrams for the 0–10 cm and 10–20 cm soil layers, respectively. (**c**,**d**) Phylum-level soil bacterial community heatmap analysis for the 0–10 cm and 10–20 cm soil layers, respectively.

**Figure 6 microorganisms-13-00759-f006:**
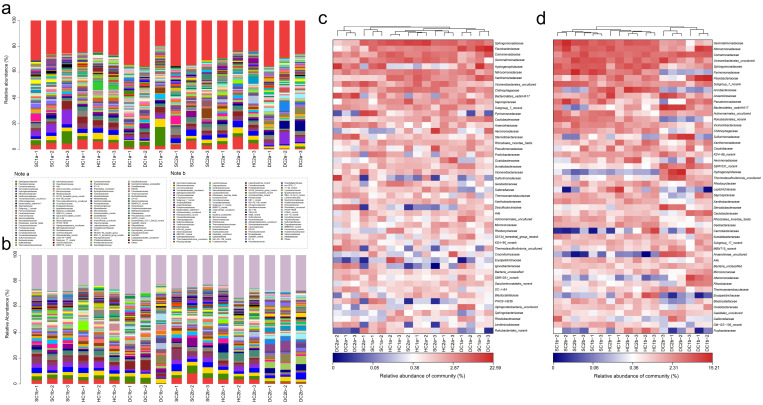
Community structure and composition analysis of soil bacteria at the family level. (**a**,**b**) Community composition of soil bacteria at the family level in the 0–10 cm and 10–20 cm soil layers, respectively; (**c**,**d**) heatmap analysis of soil bacterial communities at the family level in the 0–10 cm and 10–20 cm soil layers, respectively.

**Figure 7 microorganisms-13-00759-f007:**
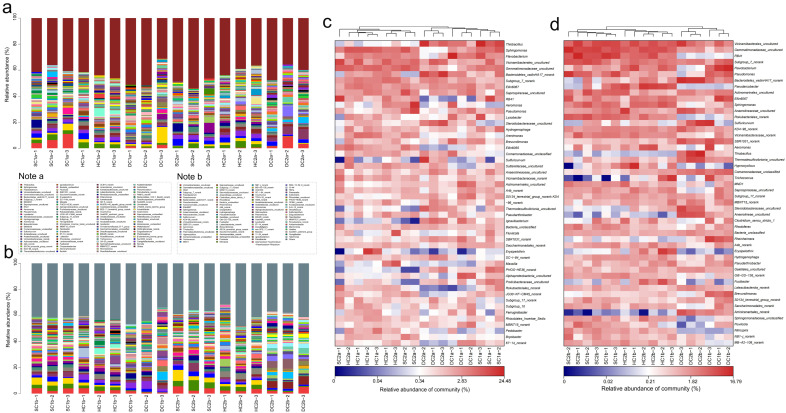
Community structure and composition analysis of soil bacteria at the genus level. (**a**,**b**) Community composition of soil bacteria at the genus level in the 0–10 cm and 10–20 cm soil layers, respectively; (**c**,**d**) heatmap analysis of soil bacterial communities at the genus level in the 0–10 cm and 10–20 cm soil layers, respectively.

**Figure 8 microorganisms-13-00759-f008:**
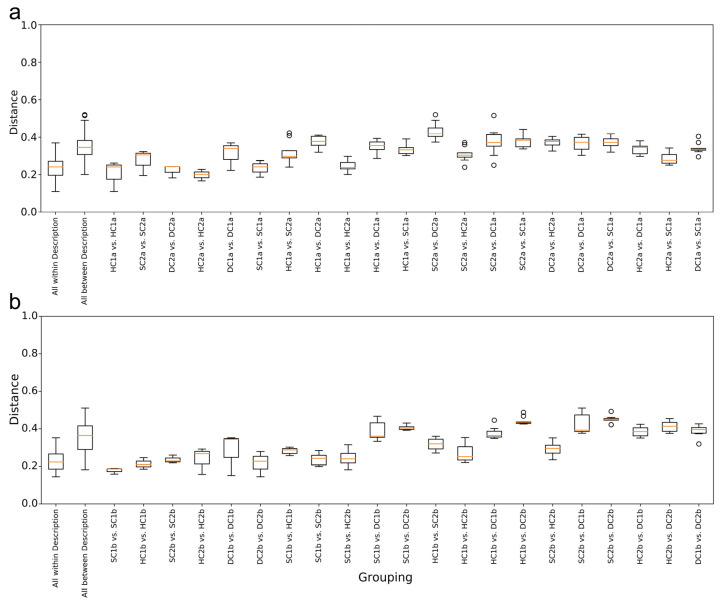
Weighted UniFrac distance analysis of bacterial communities in the Qinghai Lake estuarine and nearshore wetlands: (**a**) 0–10 cm soil layer; (**b**) 10–20 cm soil layer.

**Figure 9 microorganisms-13-00759-f009:**
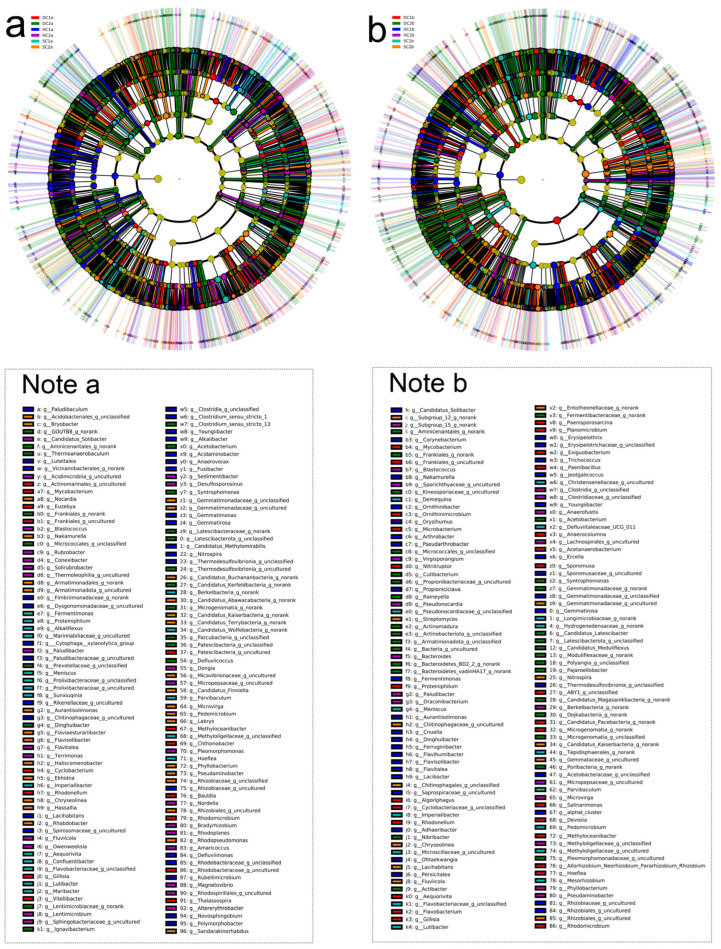
LEfSe LDA of soil bacterial communities in the estuarine wetlands and nearshore estuarine wetlands of Qinghai Lake. (**a**) Evolutionary branch diagram of LDA for the 0–10 cm soil layer; (**b**) evolutionary branch diagram of LDA for the 10–20 cm soil layer.

**Figure 10 microorganisms-13-00759-f010:**
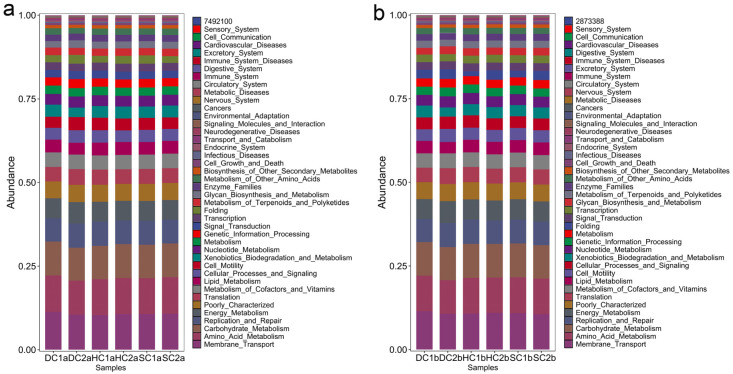
Major functional analysis of soil bacteria in the estuarine wetlands and nearshore estuarine wetlands of Qinghai Lake: (**a**) 0–10 cm soil layer; (**b**) 10–20 cm soil layer.

## Data Availability

All data generated or analyzed during this study are included in this published article.

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
