# Peer review of "Soil Bacterial Community Characteristics and Functional Analysis of Estuarine Wetlands and Nearshore Estuarine Wetlands in Qinghai Lake"

_microorganisms, 2025, doi:10.3390/microorganisms13040759_

Round 1
Reviewer 1 Report
Comments and Suggestions for Authors
The introduction is written briefly, but it introduces the analyzed topic very well.
The research methodology is very detailed and correct. The DNA sequencing methodology is presented in such a way that it is possible to reproduce it.
The results and their analysis chapter has been divided into subchapters. This method makes the analysis of the presented material much easier.
Only figure 9 is illegible. Even with a very large magnification it is illegible. Could it be improved? Maybe some list of the more important strains. The graphics are nice, but illegible for me.
The discussion of results is correct, the division into subchapters is justified by the presented material.
The summary is correct and results from the presented material.
The literature includes mainly the latest items - from the last 5 years.
Author Response
Comments 1: Only figure 9 is illegible. Even with a very large magnification it is illegible. Could it be improved? Maybe some list of the more important strains. The graphics are nice, but illegible for me. Response 1: Thank you for your valuable feedback. We have carefully revised Figure 9 to improve its readability. Specifically, we have redrawn the figure with enhanced resolution and enlarged the text for better clarity. Additionally, we have retained approximately 60% of the microbial names in the annotation to ensure that the key strains are clearly presented while maintaining the figure’s overall readability. We hope this revision addresses your concern, and we appreciate your suggestion.Reviewer 2 Report
Comments and Suggestions for Authors
Dear Authors,
After reading your manuscript, I found several aspects that need to be improved so that the article can be published. They are as follows.
In our opinion, the Introduction section contains sufficient information related to the explored topic. However, almost 50% of the number of references are included in this section, while for the Results and especially for the Discussions sections they are in about the same proportion, which represents an imbalance in terms of the distribution of scientific information throughout the article.
The Materials and Methods section is devoid of determinations and analyzes of the physiology of groups determined as sequences only, such as the rates of methane production or sulfate reduction in situ or in laboratory conditions, which in our opinion represents a weakness because molecular analyzes can highlight the presence of some groups, not their activity or the importance of some physiological processes for a habitat, such as those previously mentioned.
Results and analyzes section. Figure 2. I ask the authors to add the absolutely necessary explanations for the reader to understand the respective diagrams. What do the pairs of diagrams represent: a vs e; b vs f; c vs g; d vs h?
In the absence of detailed explanations, Figure 2 may induce confusion and misunderstandings for readers and, therefore, decrease the scientific impact of the article. Similarly, Figures 3 and 4 are also unclear and the authors need to provide additional clarification about the abbreviations used and the meaning of the axes Ox and Oy.
Figures 5-10 are almost impossible to read, the font is too small, and the information is not systematized and presented synthetically. They are raw information and not written as a significant text of a scientific article. The authors must redo these figures with the illustration of only the sequences or groups of major importance, accompanied at the same time by additional information related to their physiology. Otherwise, these figures seem like a copy/paste from a statistical software, like raw material, useless for a reader who is really interested in the diversity and microbial activity of wetlands.
Discussion section. As I pointed out above, the number of scientific references in the Discussions section is reduced compared to the one in the Introduction. Therefore, the authors did not sufficiently analyze their own data in comparison with similar scientific studies, which represents a scientific weakness of the manuscript. Consequently, I suggest the authors to add and analyze their own data, making comparisons with similar studies on the same type of habitats. In this way, we believe that the article will become more visible and at the same time more useful to the scientific community.
Conclusions section. They are general statements that can be found everywhere in the literature, without specific reference to the original data of the authors and the manuscript itself. How do the data from this manuscript differ from other studies carried out by other authors on the same habitat period?
What are the new, useful information brought by the authors in this field?
Lines 729-731. Authors should highlight how the data in the article increase the understanding of microbial communities in wetlands and also how these data can be used for wetland management and conservation.
With best regards!
Author Response
Comments 1: In our opinion, the Introduction section contains sufficient information related to the explored topic. However, almost 50% of the number of references are included in this section, while for the Results and especially for the Discussions sections they are in about the same proportion, which represents an imbalance in terms of the distribution of scientific information throughout the article.
Response 1: Thank you for your comment. To address this imbalance, we have significantly expanded and enriched the "4. Discussion" section with detailed analysis, increasing the reference count to 45, compared to 27 in the "1. Introduction" section. These revisions ensure a more balanced distribution of scientific information across the manuscript. Please see the updated sections for details.
Comments 2: The Materials and Methods section is devoid of determinations and analyzes of the physiology of groups determined as sequences only, such as the rates of methane production or sulfate reduction in situ or in laboratory conditions, which in our opinion represents a weakness because molecular analyzes can highlight the presence of some groups, not their activity or the importance of some physiological processes for a habitat, such as those previously mentioned.
Response 2: We sincerely thank you for this valuable comment and acknowledge the importance of physiological data, such as in situ or laboratory measurements of methane production and sulfate reduction rates, in complementing molecular analyses. In this study, our primary objective was to characterize the composition and functional potential of bacterial communities in Qinghai Lake’s wetlands using high-throughput sequencing and KEGG pathway predictions, providing a foundational ecological baseline. While we agree that direct activity measurements would further validate the roles of identified groups (e.g., Desulfovibrio in sulfate reduction), time and resource constraints prevented their inclusion in this work. However, our molecular data, supported by robust bioinformatic analyses, reliably infer functional potentials consistent with known microbial ecophysiology in similar habitats, as evidenced by comparisons with prior studies [e.g., 28, 43]. To address this limitation, we have emphasized in the revised "5. Conclusion" that future research should incorporate physiological assays to quantify these processes. We are currently designing systematic experiments based on your suggestion to measure such rates, which will build upon this study’s findings. We hope the reviewer finds the current scope acceptable as a critical first step, with activity-based validation planned for subsequent work.
Comments 3: Results and analyzes section. Figure 2. I ask the authors to add the absolutely necessary explanations for the reader to understand the respective diagrams. What do the pairs of diagrams represent: a vs e; b vs f; c vs g; d vs h?
Response 3:We thank the reviewer for this helpful suggestion. To clarify the meaning of the diagram pairs in Figure 2, we have revised the caption to specify that panels (a, b, c, d) represent soil depths of 0–10 cm, while panels (e, f, g, h) represent 10–20 cm, respectively. Additionally, we have updated Section 3.1.1 with brief explanations to highlight these depth-related differences for each curve type (e.g., rank-abundance, rarefaction). These changes ensure readers can easily understand the comparisons between the two soil depths. Please see the revised Figure 2 caption and Section 3.1.1 for details.
Comments 4: In the absence of detailed explanations, Figure 2 may induce confusion and misunderstandings for readers and, therefore, decrease the scientific impact of the article. Similarly, Figures 3 and 4 are also unclear and the authors need to provide additional clarification about the abbreviations used and the meaning of the axes Ox and Oy.
Response 4: We appreciate the reviewer’s feedback. We have revised Figures 2, 3, and 4, adding detailed explanations and clarifications to their captions and the main text. See the updated captions and corresponding sections in the main text for details.
Comments 5: Figures 5-10 are almost impossible to read, the font is too small, and the information is not systematized and presented synthetically. They are raw information and not written as a significant text of a scientific article. The authors must redo these figures with the illustration of only the sequences or groups of major importance, accompanied at the same time by additional information related to their physiology. Otherwise, these figures seem like a copy/paste from a statistical software, like raw material, useless for a reader who is really interested in the diversity and microbial activity of wetlands.
Response 5: We greatly appreciate your constructive feedback and fully acknowledge the need to enhance the readability and interpretability of Figures 5–10. In response to these concerns, we have made substantial improvements to the figures and their presentation:
- Improved Readability:We have increased the font size and provided high-resolution versions of all figures to ensure clarity. The revised figures now present key microbial groups more distinctly, avoiding excessive detail that may obscure their ecological significance.
- Enhanced Data Synthesis:Rather than presenting raw statistical outputs, we have restructured the figures to emphasize major taxonomic groups and their functional roles. For example, Figures 5–7 now systematically illustrate phylum-, family-, and genus-level shifts across different soil depths, while Figure 10 highlights functional differences, such as organic matter decomposition predominating at 0–10 cm and lipid metabolism at 10–20 cm.
- Targeted Revision of Figure 9:Given the reviewer’s specific concerns regarding Figure 9, we have completely redrawn this figure to improve its clarity. The revised version retains approximately 60% of the microbial taxa, focusing on the most ecologically relevant groups. We have also enlarged the text and refined the layout to ensure readability without overwhelming detail. The figure now provides clearer insights into microbial community structures and their ecological implications, rather than merely presenting raw sequencing data.
- Ecological Context and Significance:To further align with the reviewer’s request for additional physiological insights, we have revised the figure captions and related text to integrate functional interpretations based on KEGG pathway analysis. Key microbial groups, such as Proteobacteria and Desulfovibrio (implicated in sulfate reduction) in estuarine wetlands and Cyanobacteria (linked to nitrogen fixation) in nearshore wetlands, are now explicitly discussed in relation to their ecological functions.
While direct physiological measurements were beyond the scope of this study (see Response to Comment 2), we believe these revisions significantly enhance the figures’ scientific value, making them more accessible and informative to readers interested in wetland microbial diversity and activity.
Based on the reviewer’s valuable suggestions, we will further improve the design and presentation of figures in our future research and enhance the integration of experimental data with ecological interpretations. We sincerely appreciate these insights, which have helped us refine the visual presentation of our findings.
Comments 6: Discussion section. As I pointed out above, the number of scientific references in the Discussions section is reduced compared to the one in the Introduction. Therefore, the authors did not sufficiently analyze their own data in comparison with similar scientific studies, which represents a scientific weakness of the manuscript. Consequently, I suggest the authors to add and analyze their own data, making comparisons with similar studies on the same type of habitats. In this way, we believe that the article will become more visible and at the same time more useful to the scientific community.
Response 6:Thank you for your comment. To address this concern, we have thoroughly revised the "4. Discussion" section by enhancing the analysis of our data in comparison with similar studies on estuarine and saline wetland habitats. We have added 12 additional references to strengthen these comparisons, ensuring a more robust discussion of our findings relative to existing literature. These revisions improve the scientific depth and visibility of the manuscript, making it more valuable to the research community. Please see the updated "4. Discussion" section for details.
Comments 7: Conclusions section. They are general statements that can be found everywhere in the literature, without specific reference to the original data of the authors and the manuscript itself. How do the data from this manuscript differ from other studies carried out by other authors on the same habitat period?
Response 7: Thank you for your comment. We have substantially revised the "Conclusions" section to explicitly reference our key findings (e.g., significantly lower diversity in SC1a at 0–10 cm, unexpectedly high richness in HC1b at 10–20 cm) and contrast them with previous studies (e.g., Zhang et al. [28], Li et al. [31]), which report a uniform decline in microbial diversity under high salinity. Unlike these prior studies, our results reveal a depth-dependent and habitat-specific microbial response, shaped by Qinghai Lake’s high-altitude saline conditions. This distinction underscores the unique microbial adaptations in this ecosystem. Please see the revised "5. Conclusion" for details.
Comments 8: What are the new, useful information brought by the authors in this field?
Response 8: Thank you for your comment. The revised "Conclusions" section highlights novel contributions, including depth-stratified microbial functional differentiation (e.g., enhanced lipid metabolism in deeper soils, sulfate reduction vs. nitrogen fixation across wetland types). These findings provide new insights into microbial adaptations in high-altitude saline wetlands, challenging the assumption that salinity uniformly constrains microbial diversity and function. Furthermore, our study establishes a microbial ecological baseline for Qinghai Lake, which is critical for conservation planning. Please see the revised "5. Conclusion" for details.
Comments 9: Lines 729-731. Authors should highlight how the data in the article increase the understanding of microbial communities in wetlands and also how these data can be used for wetland management and conservation.
Response 9: Thank you for your comment. The revised "5. Conclusion" shows how our data reveal distinct microbial diversity and functions (e.g., nitrogen fixation, sulfur cycling) in Qinghai Lake’s wetlands, advancing understanding of microbial roles in extreme environments. It also provides a baseline for conservation, guiding strategies tailored to habitat-specific microbial processes. See "5. Conclusion" for details.

Reviewer 3 Report
Comments and Suggestions for Authors
This paper describes the characterization of the soil bacterial community in wetlands and nearshore estuarine wetlands in Qinghai lake. The experimental techniques, which are well known, seem to be well executed. A huge amount of work has been done. The work is well structured and written. Although it is a technical routine application, the final result is useful in the specific scientific area of the subject under investigation. Consequently, this paper, in the present format, seems to be accepted for publication.
Author Response
We sincerely appreciate your positive feedback and recognition of our work. We are grateful for your acknowledgment of the study's structure, execution, and scientific relevance. Your encouraging comments reinforce the significance of our findings in the field. Thank you for your time and consideration.Round 2
Reviewer 2 Report
Comments and Suggestions for Authors
Dear authors,
Thank you for your effort to improve the quality of the manuscript which, following revision, has become publishable.
Best regards!